# Long-timescale anti-directional rotation in *Drosophila* optomotor behavior

Omer Mano[1], Minseung Choi[2], Ryosuke Tanaka[3], Matthew S Creamer[3], Natalia CB Matos[3], Joseph W Shomar[4], Bara A Badwan[5], Thomas R Clandinin[2], Damon A Clark[1,3,4,6]*

[1]Department of Molecular, Cellular, and Developmental Biology, Yale University, New Haven, United States; [2]Department of Neurobiology, Stanford University, Stanford, United States; [3]Interdepartmental Neuroscience Program, Yale University, New Haven, United States; [4]Department of Physics, Yale University, New Haven, United States; [5]Department of Chemical Engineering, Yale University, New Haven, United States; [6]Department of Neuroscience, Yale University, New Haven, United States

*For correspondence:
damon.clark@yale.edu

**Abstract** Locomotor movements cause visual images to be displaced across the eye, a retinal slip that is counteracted by stabilizing reflexes in many animals. In insects, optomotor turning causes the animal to turn in the direction of rotating visual stimuli, thereby reducing retinal slip and stabilizing trajectories through the world. This behavior has formed the basis for extensive dissections of motion vision. Here, we report that under certain stimulus conditions, two *Drosophila* species, including the widely studied *Drosophila melanogaster*, can suppress and even reverse the optomotor turning response over several seconds. Such 'anti-directional turning' is most strongly evoked by long-lasting, high-contrast, slow-moving visual stimuli that are distinct from those that promote syn-directional optomotor turning. Anti-directional turning, like the syn-directional optomotor response, requires the local motion detecting neurons T4 and T5. A subset of lobula plate tangential cells, CH cells, show involvement in these responses. Imaging from a variety of direction-selective cells in the lobula plate shows no evidence of dynamics that match the behavior, suggesting that the observed inversion in turning direction emerges downstream of the lobula plate. Further, anti-directional turning declines with age and exposure to light. These results show that *Drosophila* optomotor turning behaviors contain rich, stimulus-dependent dynamics that are inconsistent with simple reflexive stabilization responses.

## Editor's evaluation

The present study provides a valuable new perspective on the optomotor response based on an inversion of the behavior under specific (non-natural) conditions that may help elucidate the principles of this specific behavior. The evidence provided is convincing.

## Introduction

Visual navigation requires active mechanisms to stabilize locomotor trajectories through the world. Insects exhibit an optomotor turning response, a behavior in which they rotate their bodies in the direction of visual patterns that rotate about them (*Hassenstein and Reichardt, 1956*; *Götz and Wenking, 1973*; *Buchner, 1976*). This behavior is analogous to optomotor turning responses in fish (*Clark, 1981*) and the optokinetic response in mammals (*Koerner and Schiller, 1972*). In insects, this response is thought to be a course stabilizing mechanism that minimizes retinal slip, allowing animals to maintain their trajectory in the face of external or unexpected rotational forces (*Götz and*

**Figure 1.** Flies turn opposite to the stimulus direction in high-contrast conditions. (**a**) We measured fly turning behavior as they walked on an air-suspended ball. Stimuli were presented over 270° around the fly.

*Figure 1 continued on next page*

*Figure 1 continued*

(**b**) We presented drifting sinusoidal gratings for 5 s (shaded region) with either high contrast (c=1.0) or low contrast (c=0.25). When high-contrast sinusoidal gratings were presented, flies initially turned in the same direction as the stimulus, then started turning in the opposite direction after ~1 s of stimulation. Under low-contrast conditions, flies turned continuously in the same direction as the stimulus. In these experiments, the sine waves had a wavelength of 60° and a temporal frequency of 1 Hz. Shaded patches represent ±1 SEM. N=10 flies. (**c**) We swept contrast between 0 and 1 and measured the mean turning response during the first 0.5 s (purple, purple bar in **b**) and during the last 4 s of the stimulus (brown, brown line in **b**). The response in the first 0.5 s increased with increasing contrast, while the response in the last 4 s increased from c=0 to c=0.25, and then decreased with increasing contrast, until flies turned in the direction opposite the stimulus direction at the highest contrasts. N=20 flies. (**d**) We repeated the presentation of drifting sinusoidal gratings, this time in the lab of author TRC, using a similar behavioral apparatus. Stimulus parameters were as described in (**b**). In these experiments, the population average shows that flies proceeded to zero net turning at high contrasts, but some individual flies exhibited anti-directional turning responses. N=20 flies. (**e**) We repeated the experiments with *D. yakuba*, also in the lab of TRC, and observed that this species exhibited a robust anti-directional turning response to high-contrast gratings and a classical syn-directional turning response to low-contrast gratings. N=11 flies.

The online version of this article includes the following figure supplement(s) for figure 1:

**Figure supplement 1.** Individual *D. melanogaster* flies in TRC lab experiments show anti-directional turning.

**Figure supplement 2.** Flies perform anti-directional turning under a wide range of stimulus and growing conditions.

*Wenking, 1973*; *Götz, 1975*). For instance, if an insect attempts to walk in a straight line, it may slip and turn to the right. From the point of view of the insect, this turn is observed as optic flow rotating to the left. By responding to this leftward optic flow with a leftward turn, the insect can recover its original trajectory.

In fruit flies, the optomotor response relies on well-characterized circuitry (*Yang and Clandinin, 2018*). Photoreceptor signals are split into parallel ON and OFF pathways in the lamina and medulla (*Joesch et al., 2010*; *Clark et al., 2011*; *Behnia et al., 2014*; *Strother et al., 2014*), which are not direction-selective. These signals provide input to T4 and T5 cells, which compute direction-selective responses along four directions at every point in the fly visual field (*Bausenwein et al., 1992*;

*Maisak et al., 2013*; *Takemura et al., 2013*; *Shinomiya et al., 2019*; *Henning et al., 2022*). The outputs of T4 and T5 cells are then summed across visual space by lobula plate tangential cells (LPTCs) (*Joesch et al., 2008*; *Schnell et al., 2012*; *Maisak et al., 2013*; *Mauss et al., 2015*; *Barnhart et al., 2018*). Different LPTCs provide distinct signals about the overall pattern of motion surrounding the fly, and have been linked to head and body movements (*Krapp and Hengstenberg, 1996*; *Haikala et al., 2013*; *Kim et al., 2017*).

There have been several reports of flies turning in the direction opposite to what is predicted by the optomotor turning response. In some cases, these counter-intuitive behaviors were observed using periodic stimuli with spatial wavelengths smaller than the receptive field of individual ommatidia, and thus can be accounted for by aliasing (*Götz, 1964*; *Götz, 1970*; *Buchner, 1976*). Work in a tethered flight simulator showed that when a moving pattern is presented in front of the fly, the animal turned in the direction of the stimulus motion (*Tammero et al., 2004*), as expected (*Götz, 1968*). However, if the moving pattern was presented behind the fly, it attempted to turn in the direction opposite to stimulus motion (*Tammero et al., 2004*). In a different experimental preparation, rotational patterns were presented on a dome around freely walking flies (*Williamson et al., 2018*). Under these conditions, flies generally turned in the direction of motion of the stimulus, but these rotations were often punctuated by brief, large-magnitude saccades in the opposite direction. Similarly, experiments using flight simulators have reported spikes in the torque in the direction opposite the stimulus rotation (*Wolf and Heisenberg, 1990*). Interestingly, zebrafish have also been observed to turn in the opposite direction of optic flow under certain conditions (*Bak-Coleman et al., 2015*).

Here, we show that rotational stimuli can elicit strong, consistent anti-directional turning behavior in two drosophilid species, *Drosophila melanogaster* and *Drosophila yakuba*. We report that flies respond to high-contrast, high-luminance rotational motion stimuli by first turning in the direction of stimulus motion, and then reversing their trajectory after approximately 1 s, depending on the species. In *D. melanogaster*, we characterize the dynamics of this behavior and the stimuli that drive it, showing that it is distinct from prior observations of anti-directional turning. The behavior depends critically on adaptation to back-to-front motion. We use the genetic tools available in *D. melanogaster* to show that this behavior relies on the motion detecting neurons T4 and T5. Silencing horizontal system (HS) neurons and CH neurons, two wide-field neurons downstream of T4 and T5, resulted in small changes in this complex turning behavior. However, the visually evoked responses of these direction-selective neurons could not account for the anti-directional behavior. Thus, the observed reversal must be mediated by downstream circuitry. Overall, these results show that circuits in the fly generate behaviors that oppose the direction of wide-field visual motion, showing that *Drosophila* turning responses are more complex than a simple stabilizing reflex.

## Results

### Anti-directional turning responses to high-contrast stimuli

Optomotor turning responses are central to gaze stabilization, so we sought to examine this response across different conditions. Many studies have investigated this behavior using stimuli with low contrast, low light intensity, or both (*Götz and Wenking, 1973*; *Buchner, 1976*; *Rister et al., 2007*; *Seelig et al., 2010*; *Bahl et al., 2013*; *Bosch et al., 2015*), at a variety of different speeds. However, natural scenes can have relatively high contrast and luminance, and such conditions have been poorly explored in the laboratory. In this experiment, we presented flies with rotational stimuli using high contrast and relatively high luminance.

We tethered individual female *D. melanogaster* above a freely rotating ball to characterize the optomotor response (*Buchner, 1976*; *Creamer et al., 2019*; *Figure 1a*). As expected, low-contrast, slow-moving sinusoidal gratings caused flies to turn in the same direction as the moving gratings via the classical optomotor turning response (*Figure 1b*; *Hassenstein and Reichardt, 1956*; *Götz, 1964*; *Buchner, 1976*; *Tammero et al., 2004*; *Seelig et al., 2010*; *Clark et al., 2011*; *Bahl et al., 2013*; *Silies et al., 2013*; *Clark et al., 2014*; *Bahl et al., 2015*; *Leonhardt et al., 2016*; *Salazar-Gatzimas et al., 2016*; *Strother et al., 2017*; *Creamer et al., 2018*; *Strother et al., 2018*). However, when we changed the stimulus to high-contrast sinusoidal gratings (nominal 100% Weber contrast), flies turned in the stimulus direction for approximately 1 s, but then reversed course, and turned in the direction opposite to the stimulus motion for the duration of the stimulus presentation. Because this turning

response is in the opposite direction of stimulus and the syn-directional optomotor turning response, we refer to it as anti-directional turning.

We swept a range of contrasts and compared the fly turning in the first 500 ms to the turning after 1 s (*Figure 1c*). As contrast increased, the flies turned faster during the first half second of stimulus presentation, reaching a plateau at around 0.5 contrast, consistent with previous results (*McCann and MacGinitie, 1965*; *Buchner, 1976*; *Heisenberg and Buchner, 1977*; *Duistermars et al., 2007*; *Bahl et al., 2015*; *Strother et al., 2017*). Fly behavior after the first second of stimulation was more complex. At contrasts between 0 and 0.25, flies turned in the same direction as the stimulus, with faster turning as the contrast increased. When the contrast was greater than 0.25, turning decreased, lowering to no net sustained turning at around 0.8 contrast. Above a contrast of 0.8, flies began to turn in the direction opposite the stimulus.

These initial experiments took place in the lab of author DAC. To confirm that these unexpected responses did not reflect some idiosyncrasy of one specific behavioral apparatus or environment, we repeated these experiments in a second lab, that of author TRC. Under similar conditions, using the same strain of *D. melanogaster*, we reproduced the rapid deceleration after an initial, transient syn-directional response (*Figure 1d*), with some individual flies exhibiting significant anti-directional turning (*Figure 1—figure supplement 1*). This demonstrates that the key features of this behavioral response are stable across experimental systems and laboratories, though the magnitude of anti-directional turning behavior in *D. melanogaster* is sensitive to some unknown experimental parameter differences between the laboratories.

Individual strains of *D. melanogaster*, and other drosophilid species, display significant variation in their locomotor patterns during walking (*York et al., 2022*). Indeed, when we tested a Canton-S *D. melanogaster* strain, we observed milder but significant anti-directional turning at long timescales (*Figure 1—figure supplement 2b*). We reasoned that a strong test of the generality of anti-directional turning would be to examine turning behavior in another species, and selected *D. yakuba*. Strikingly, *D. yakuba* also displayed anti-directional turning behavior under similar conditions (*Figure 1e*). Thus, this behavior is not an idiosyncratic feature of a single laboratory strain.

## Conditions for anti-directional turning behaviors

While anti-directional turning behaviors have been reported before, other groups have presented similar stimuli without observing anti-directional behavior (*Götz and Wenking, 1973*; *Buchner, 1976*; *Rister et al., 2007*; *Seelig et al., 2010*; *Bahl et al., 2013*; *Bosch et al., 2015*). We wondered what aspects of our experimental setup could lead to these behavioral differences. In our experiments, anti-directional turning was strongly linked to display brightness (*Figure 1—figure supplement 2a*). When the mean brightness of the screens was reduced from 100 cd/m$^2$ to 1 cd/m$^2$, we saw no anti-directional turning in 5 s trials (though average optomotor behavior did decrease over the course of the stimulus presentation). When we further reduced the mean brightness to 0.1 cd/m$^2$, flies persisted in their optomotor behavior throughout the stimulus presentation. We note that in these low-luminance experiments, low levels of ambient light in the nominally dark experimental rig could also reduce the effective contrast of the stimulus.

We tested a variety of other factors that might affect anti-directional turning. Anti-directional turning occurred when experiments were run both at hot temperatures and at room temperature (*Figure 1—figure supplement 2b*). We also observed anti-directional behavior when flies were reared in the dark and on different media. We also tested several other experiment conditions (*Figure 1—figure supplement 2c*). Flies responded with anti-directional turning to high-contrast stimuli presented at both blue and green wavelengths. We glued fly heads to their thorax to ensure stimuli could not be affected by head movements (*Haikala et al., 2013*; *Kim et al., 2017*), but found no difference between head-fixed and head-free flies. We did find a few factors that modulated anti-directional turning behavior. In particular, rearing *D. melanogaster* at 25°C instead of 20°C or testing flies that were 2 weeks old instead of 12–60 hr old both reduced overall turning behavior and eliminated anti-directional turning. In these cases, optomotor turning still decreased over the course of the 5 s, high-contrast trials, but did not reverse. As details of rearing temperature and the age at which behavior tests are run often vary across labs, it is possible that these factors, as well as stimulus brightness, account for the differences between our observations and the previous literature.

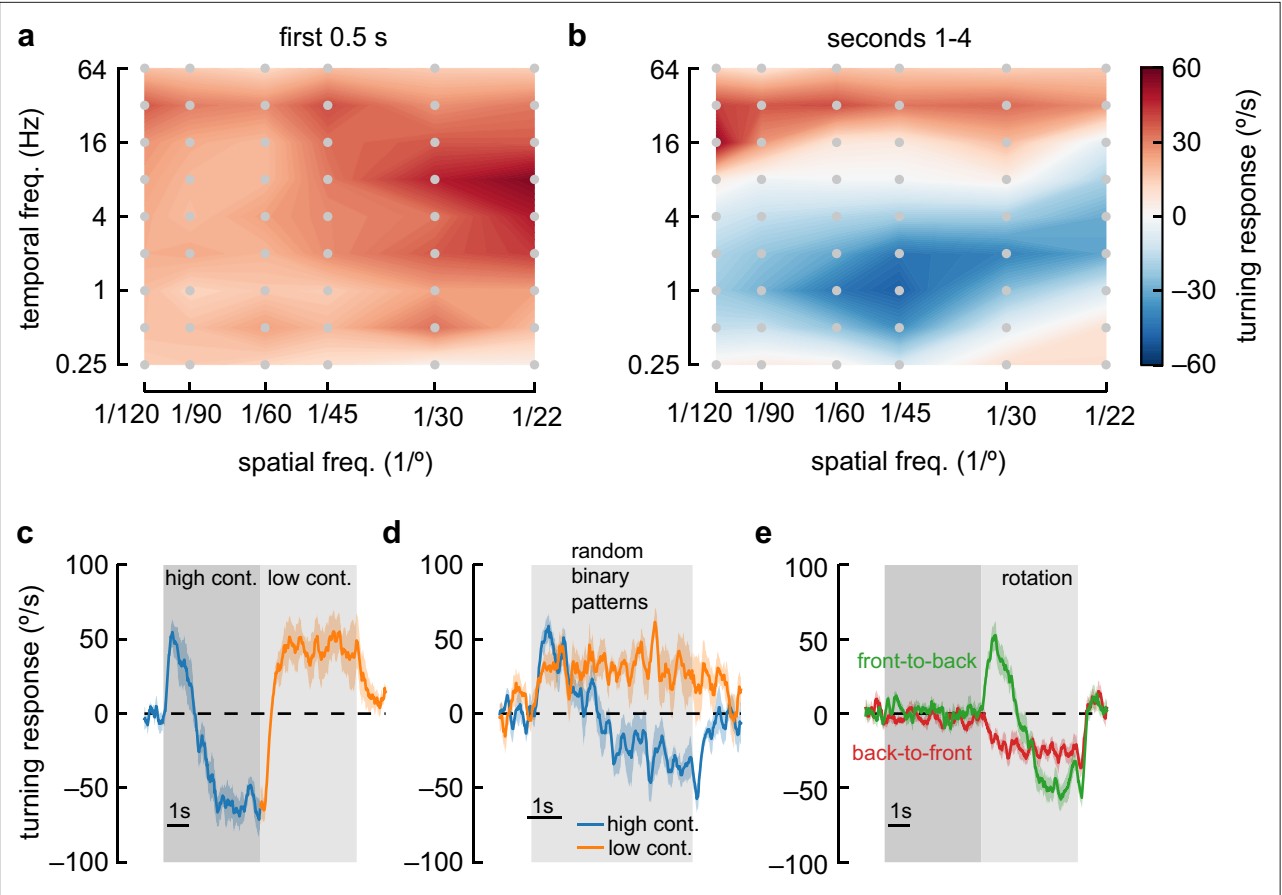

**Figure 2.** Anti-directional turning behavior has distinct tuning and is driven by adaptation. (**a**) Heatmap of fly turning velocity during the first 0.5 s of sinusoidal grating stimulation under high-contrast conditions and variable temporal and spatial frequencies. The flies turned in the direction of the stimulus across all conditions and responded most to 8 Hz, 22° stimuli. N=16, 21, 17, 21, 7, and 22 flies for spatial frequencies 1/120°, 1/90°, 1/60°, 1/45°, 1/30°, and 1/22° respectively. (**b**) Heatmap as in (**a**), measured during the last 4 s of stimulation. Flies turned in the same direction as the stimulus at high and low temporal frequencies, but in the opposite direction of the stimulus at intermediate temporal frequencies, with a maximal anti-directional response at wavelengths between 30° and 60°. (**c**) Switching stimulus contrast from high to low after 5 s caused flies to revert to syn-directional behavior after the anti-directional response. N=7 flies. (**d**) Presenting rotating random binary patterns (5° vertical strips rotating at 150 °/s) induced anti-directional turning similar to that elicited by rotating sine wave gratings. N=7 flies. (**e**) We presented flies with 5 s of 'translational' stimuli (dark shaded region), with high-contrast sinusoidal gratings moving either front-to-back or back-to-front, bilaterally, for 5 s. After that, we presented high-contrast rotational sinusoidal grating stimuli (60° wavelength, 1 Hz). Front-to-back stimulation did not affect the subsequent response to rotational stimuli, but back-to-front stimuli caused flies to turn immediately in the opposite direction of the stimulus. N=18 flies.

The online version of this article includes the following figure supplement(s) for figure 2:

**Figure supplement 1.** As in *Figure 2e*, with different adapters.

## Distinct spatiotemporal tuning of the anti-directional behavioral response

To further characterize the anti-directional response, we swept the spatial and temporal frequency of the sinusoidal grating stimulus. Using only Weber contrasts of 1, we compared the early response (first quarter second, *Figure 2a*) to the late response (after 1 s, *Figure 2b*). *D. melanogaster* always turned in the optomotor direction during the early stimulus response. In this early response, flies turned most vigorously to stimuli with short spatial frequencies (~20° wavelength) and fast temporal frequencies (~8 Hz), in agreement with earlier studies (*Tammero et al., 2004*; *Creamer et al., 2018*; *Strother et al., 2018*). However, during the longer-timescale response to high-contrast stimuli, flies only turned in the optomotor direction at very high temporal frequencies (>~16 Hz) and at very low temporal frequencies (<0.5 Hz). At intermediate temporal frequencies, flies showed a sustained anti-directional response. The maximal anti-directional response was achieved at 1 Hz and 45° wavelength,

distinct from the conditions for peak classical turning responses. Interestingly, the stimuli that elicit the strongest anti-directional response appear similar to those that maximally activate T4 and T5 neurons when those neurons are measured in head-fixed flies (*Maisak et al., 2013*; *Leong et al., 2016*; *Arenz et al., 2017*; *Creamer et al., 2018*; *Strother et al., 2018*; *Wienecke et al., 2018*).

## Anti-directional turning results from adaptation effects

We were intrigued by the switch from syn-directional to anti-directional turning behavior. To investigate the dynamics of these changes, we presented a rotating sinusoidal stimulus at contrast 1 for 5 s, and then changed the contrast to 0.25 (*Figure 2c*). After the switch to low contrast, the flies quickly reverted classical, syn-directional optomotor behavior, demonstrating that no long-term switch in directional turning occurs during high-contrast stimulus presentation. This effect did not depend on the periodic nature of these stimuli: a rotating stimulus consisting of 5°-wide vertical bars with randomly chosen, binary contrasts (*Clark et al., 2014*) yielded similar behavioral responses (*Figure 2d*).

To further isolate the causes of this switch in behavior, we developed a protocol to adapt the fly to different stimuli before presenting the high-contrast rotational sinusoidal gratings to elicit the anti-directional turning response. This adapting protocol consisted of 5 s presentations of an adaptor stimulus, followed by a high-contrast rotational stimulus (*Figure 2e*, *Figure 2—figure supplement 1*). The adaptor stimuli we tested were a uniform gray screen, a stationary high-contrast sinusoid, a closed-loop high-contrast sinusoid, or a high-contrast 'translational' sinusoidal stimulus. During the closed-loop adapter, the stimulus position was yoked to the fly's own turning to better simulate real rotation, a situation we hypothesized might yield less dependence on stimulus contrast (following *Leonhardt et al., 2016*). The translational stimulus had both left and right hemifields moving either front-to-back or back-to-front across the fly's two eyes (*Creamer et al., 2018*). These translational stimuli resulted in no net turning (*Silies et al., 2013*; *Creamer et al., 2018*), but specifically adapt motion detectors in both eyes selective for front-to-back or back-to-front motion. We found that adapting with a stationary or closed-loop high-contrast sinusoid had little effect on the behavior (*Figure 2—figure supplement 1*). Similarly, adapting the fly with front-to-back stimuli did not have a strong effect on the subsequent response to rotational stimuli. However, adapting with back-to-front stimuli generated responses that no longer showed an initial syn-directional turning response, but instead exhibited anti-directional turning immediately after the rotational stimulus began. This result indicates that the anti-directional turning results from slow-timescale changes that depend on strong back-to-front motion.

## Anti-directional turning is elicited when stimuli are presented in front of the fly

A previous report of anti-directional turning behavior in flying tethered flies showed that flies turn in the opposite direction to stimuli that are presented behind their midline (*Tammero et al., 2004*). To test whether our results were caused by this effect, we split our stimulus into three regions: 90° in front of the fly, 45° in front of the midline on either side of the fly, and 45° behind the midline on either side of the fly (*Figure 3a*). We found that flies displayed anti-directional turning when presented with stimuli only in the front region or only just in front of the midline (*Figure 3b, c*). They did not display anti-directional turning when moving stimuli were presented behind the midline (*Figure 3b, c*). This suggests a different mechanism from the behaviors that depend on posterior spatial location to elicit reverse turning (*Tammero et al., 2004*).

## Anti-directional responses do not depend on saccades

Anti-directional saccades have been reported in walking and flying flies (*Wolf and Heisenberg, 1990*; *Williamson et al., 2018*). In walking flies (*Williamson et al., 2018*), flies largely turned in syn-directionally, but these turns were sometimes interrupted by brief, high-amplitude saccades in the opposite direction, against the stimulus direction. If such saccades were frequent or high amplitude, the net effect could shift the average turning we measured, creating apparent anti-directional turning. To investigate this possibility, we plotted the turning response on a per-trial basis (*Figure 3d*). We then discarded information about the magnitude of the turns and considered only the direction of the turning at each point in time (*Figure 3e*). Strikingly, in many trials, flies continued to turn opposite to the stimulus for several seconds, a behavior unlike brief saccades. This prolonged turning against

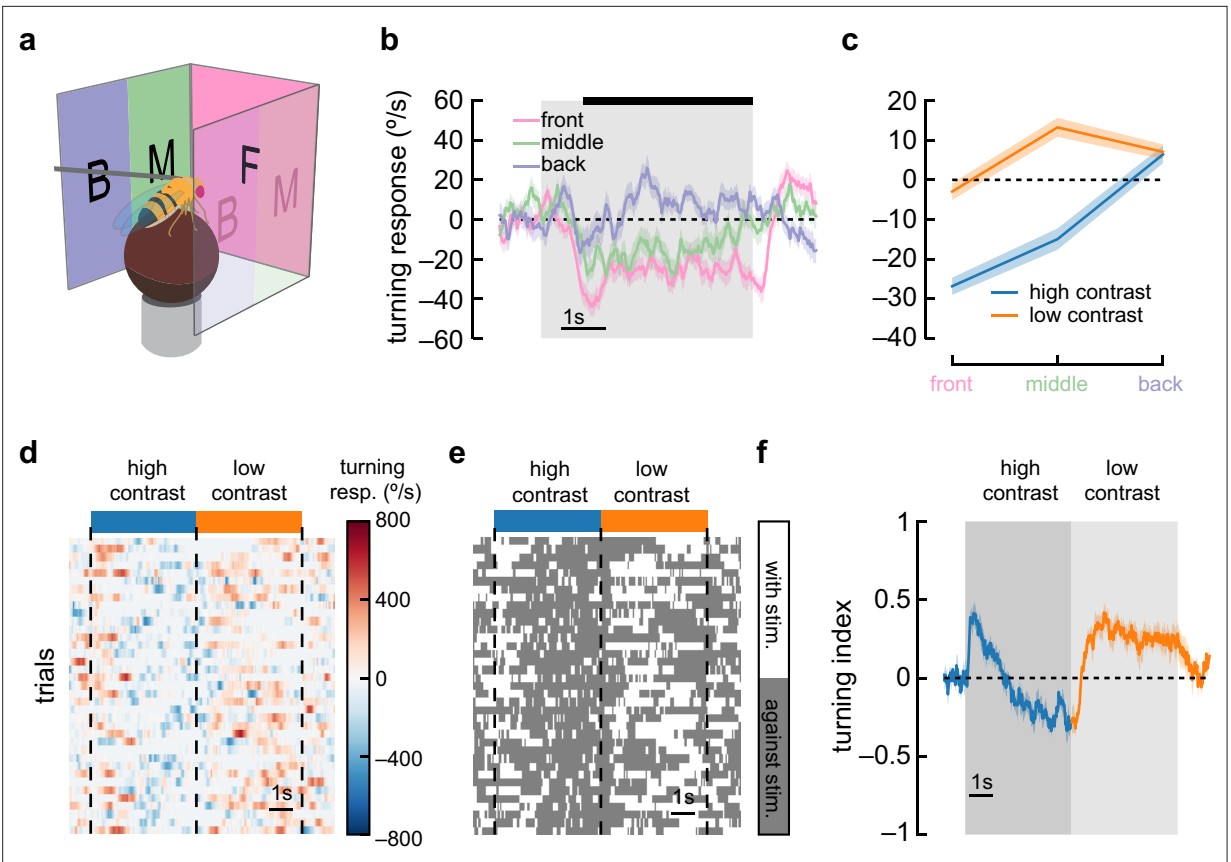

**Figure 3.** Anti-directional turning is driven by stimuli in the forward-facing visual field and is not driven by saccades. (**a**) We divided our panoramic display into three sections: the front 90°, the 45° behind the fly on either side, and a middle 45°. (**b**) High-contrast sinusoidal gratings were presented on each of these three display sections, with the remaining sections blank. Flies turned syn-directionally when stimuli were presented behind the fly, and turned anti-directionally when stimuli were presented in front of the fly. Shaded patches represent ±1 SEM. N=55 flies. (**c**) Average turning in the last 4 s of the stimulus (black bar in **b**), in low-contrast and high-contrast conditions. Shaded patches in the time trace plots represent ±1 SEM. N=55 flies. (**d**) A single fly responds to many trials of sinusoidal grating stimuli at high contrast (blue bar) and low contrast (orange bar). We show a heatmap of the fly's responses over time (horizontal axis) and across trials (vertical axis). (**e**) We can ignore the magnitude of the turning and instead only quantify whether the fly was turning in the same direction as the stimulus (white area) or in the opposite direction (dark gray area). This shows sustained anti-directional turning, not brief saccades. (**f**) Averaging the direction (but not magnitude) of turning across trials and across flies yields a turning index for each point in time. Shaded patches in the time trace plots represent ±1 SEM. N=7 flies.

The online version of this article includes the following figure supplement(s) for figure 3:

**Figure supplement 1.** Distribution of turning responses over trials and over flies.

the stimulus was also observable in the distribution of turning over time in all trials over all flies (*Figure 3—figure supplement 1*). Next, we calculated a turning index for each response timepoint (sampled at 60 Hz). This turning index represents the fraction of trials in which the fly turned in the direction of the stimulus at each timepoint minus the fraction of trials in which the fly turned in the opposite direction (*Figure 3f*). Since this turning index does not include the magnitude of turning, it is strongly affected by sustained low-amplitude turns and discounts any brief high-amplitude saccades. When presented with high-contrast stimuli, flies maintained a negative turning index, indicating that sustained turns, and not high-velocity saccades, underlie this anti-directional turning behavior. As such, it appears distinct from the reports of anti-directional saccades.

## Anti-directional turning requires elementary motion detectors

What neurons are involved in this anti-directional turning behavior? Previous work demonstrated that T4 and T5 are required for directional neural responses (*Schnell et al., 2012*), as well as for optomotor turning (*Maisak et al., 2013*; *Salazar-Gatzimas et al., 2016*; *Salazar-Gatzimas et al., 2018*), for walking speed regulation (*Creamer et al., 2018*), and for responses to visual looming stimuli (*Schilling*

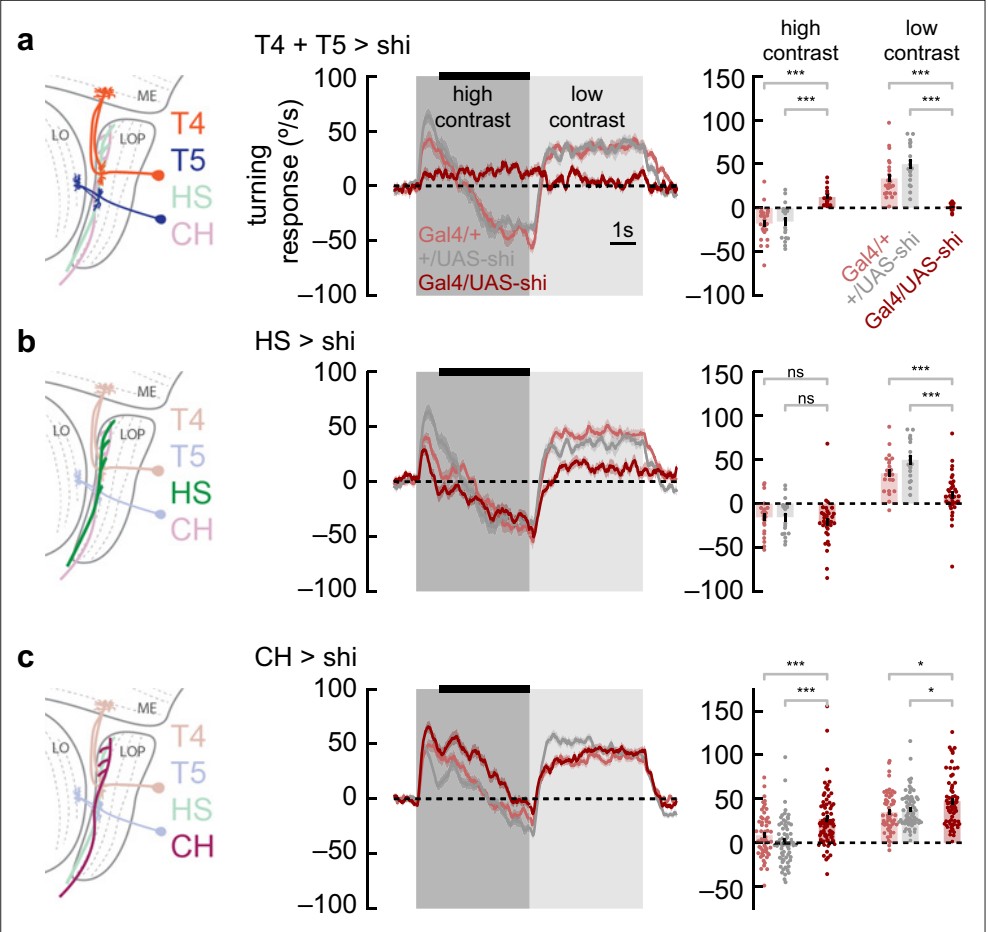

**Figure 4.** Syn-directional and anti-directional turning share common circuitry. (**a**) We silenced T4 and T5 neurons by expressing shibire[ts] selectively in those neurons. We measured turning behavior during a contrast-switching stimulus (as in *Figure 2c*). Results from flies with T4 and T5 silenced shown in dark red, while controls are in light red and gray. Average fly behavior during the last 4 s of the first contrast (black bar on left) shown as bars on the right, with individual fly behavior shown as dots. Note that the data labeled 'low contrast' are from experiments in which the low-contrast stimulus was shown before the high-contrast stimulus. Shaded patches in the time trace plots represent ±1 SEM, as do vertical lines on bar plots. *** indicates experimental results are significantly different from results, $p<0.001$ via a two-sample Student's t-test. * indicates $p<0.05$. N=17, 24, 19 flies with genotypes T4T5/Shibire[ts], T4T5/+, +/Shibire[ts]. (**b**) Results from horizontal system (HS) neuron silencing as in (**a**). Silencing HS neurons reduced syn-directional turning behavior ($p<0.001$) but did not have a strong effect on anti-directional turning. N=34, 21, 19 flies with genotypes HS/Shibire[ts], HS/+, +/Shibire[ts]. (**c**) Results from CH neuron silencing as in (**a**). CH neuron silencing reduced the degree of anti-directional turning ($p<0.001$). N=63, 57, 70 flies with genotypes CH/Shibire[ts], CH/+, +/Shibire[ts].

*and Borst, 2015*). We silenced the neurons T4 and T5 using shibire[ts] (*Kitamoto, 2001*) and measured responses to sinusoidal stimuli that switched from high to low contrast (*Figure 4a*). Flies in which T4 and T5 had been silenced displayed only minimal responses to motion stimuli, with anti-directional turning suppressed along with classical syn-directional turning. Thus, we conclude that, like optomotor turning behaviors, this anti-directional behavior depends critically on signals from T4 and T5.

## Anti-directional turning requires the CH LPTC

Since the switch from optomotor to anti-directional behavior seems to be dependent on the direction of motion adaptation (*Figure 2e*), we reasoned that neurons involved in this behavior were likely to be downstream from T4 and T5. Relatively little is known about circuitry that connects the neurons T4 and T5 to optomotor turning behavior. However, horizontal system (HS) cells are well-studied postsynaptic partners of T4 and T5 (*Joesch et al., 2008*; *Joesch et al., 2010*). These LPTCs integrate information

from front-to-back and back-to-front selective T4 and T5 cells across the fly's visual field (*Mauss et al., 2015*). HS cells have been implicated in visually evoked head turns (*Kim et al., 2017*) and body rotations in flight (*Haikala et al., 2013*) and in maintenance of direction during walking (*Fujiwara et al., 2022*). When we silenced HS neurons, we found small deficits in syn-directional turning behavior, consistent with prior results, but no deficits in anti-directional turning (*Figure 4b*), indicating that HS cells synaptic output is not required specifically for anti-directional turning behavior.

Next, we turned to the CH LPTCs. These cells are GABAergic and are both pre-synaptic and post-synaptic in the lobula plate (*Wei et al., 2020*). In blowflies, these neurons play an inhibitory role in an interconnected LPTC circuit that shapes behavior (*Borst and Weber, 2011*). When we silenced CH neurons, we found a small increase in syn-directional turning and a decrease in anti-directional turning (*Figure 4c*). Overall, silencing this neuron type caused the flies to turn more in the direction of motion. This result suggests that CH activity contributes to the anti-directional turning response. However, since adapting to back-to-front translational stimuli significantly affected the dynamics of anti-directional turning, it seems likely that other neurons beyond HS and CH neurons are involved, since these two neurons both respond selectively to front-to-back motion (*Eckert and Dvorak, 1983*; *Joesch et al., 2008*).

## Early direction-selective cells do not adapt to the stimulus

The anti-directional turning response is preceded by an initial syn-directional response. This change in behavior must be the result of changes in neural activity, but this change could happen at any point along the neural pathway between photoreceptors and motor neurons. In order to constrain possible mechanisms for generating the anti-directional turning behaviors, we used calcium imaging to interrogate the activity of direction-selective neurons during high- and low-contrast stimulation (*Figure 5a*). However, as calcium imaging experiments using two-photon microscopy require additional spectral filtering of the projector, we first confirmed that these spectral differences did not alter anti-directional turning responses. To do this, we re-measured the anti-directional turning behavior using optical filtering matched to the conditions needed for imaging. Using this spectrally distinct illuminant, we observed both syn-directional and anti-directional turning behaviors, following the previously observed dynamics (*Figure 5—figure supplement 1*).

As T4 and T5 neurons play a critical role in both the syn- and anti-directional turning responses, we first measured the calcium activity of these neurons as they responded to sine wave gratings at a range of contrasts in their preferred and null directions. The T4 and T5 neurons responded to sine wave gratings in their preferred direction by increasing their calcium activity for the full duration of the stimulus presentation, reaching a plateau after approximately 1 s (*Figure 5b, c*, middle). As we increased the contrast of the preferred direction stimuli, we found that both T4 and T5 cells had increased calcium activity throughout the contrast range (*Figure 5b, c*, right), consistent with prior measurements (*Maisak et al., 2013*). Thus, the responses of T4 and T5 cells do not capture the transition from syn-directional to anti-directional turning behavior.

Next we examined two LPTCs downstream of T4 and T5 cells. Calcium activity in HS cells followed similar trends to T4 and T5. Calcium signals increased at the start of preferred direction stimuli presentation and stayed high until the end of the presentation (*Figure 5d*, middle). Increasing contrast caused stronger calcium responses with a mild saturation effect at high contrast (*Figure 5d*, right), consistent with prior voltage measurements (*Joesch et al., 2008*). These results indicate that the changes in the time course of optomotor behavior at high contrast are not related to changes in HS neuron activity. Finally, we measured calcium activity in CH cells. CH cells responded to visual stimuli more quickly than HS cells (*Figure 5e*, middle), and showed decreased calcium signals in response to null direction stimuli (*Figure 5e*, right). However, they also showed sustained responses to high-contrast stimuli, as in T4, T5, and HS neurons. These measurements suggest that the switch from syn- to anti-directional turning behavior is driven by cells downstream of or parallel to T4, T5, HS, and CH neurons.

## Adult plasticity in anti-directional turning behavior

In behaving flies, the strength of anti-directional turning was dependent both on rearing temperature, which alters the rate of growth, and on age (*Figure 1—figure supplement 2*). This raises the possibility that syn- and anti-directional turning responses might be plastic during the early adult stages of development. To probe this possibility, we presented 1 Hz, high-contrast, rotating sinusoidal grating

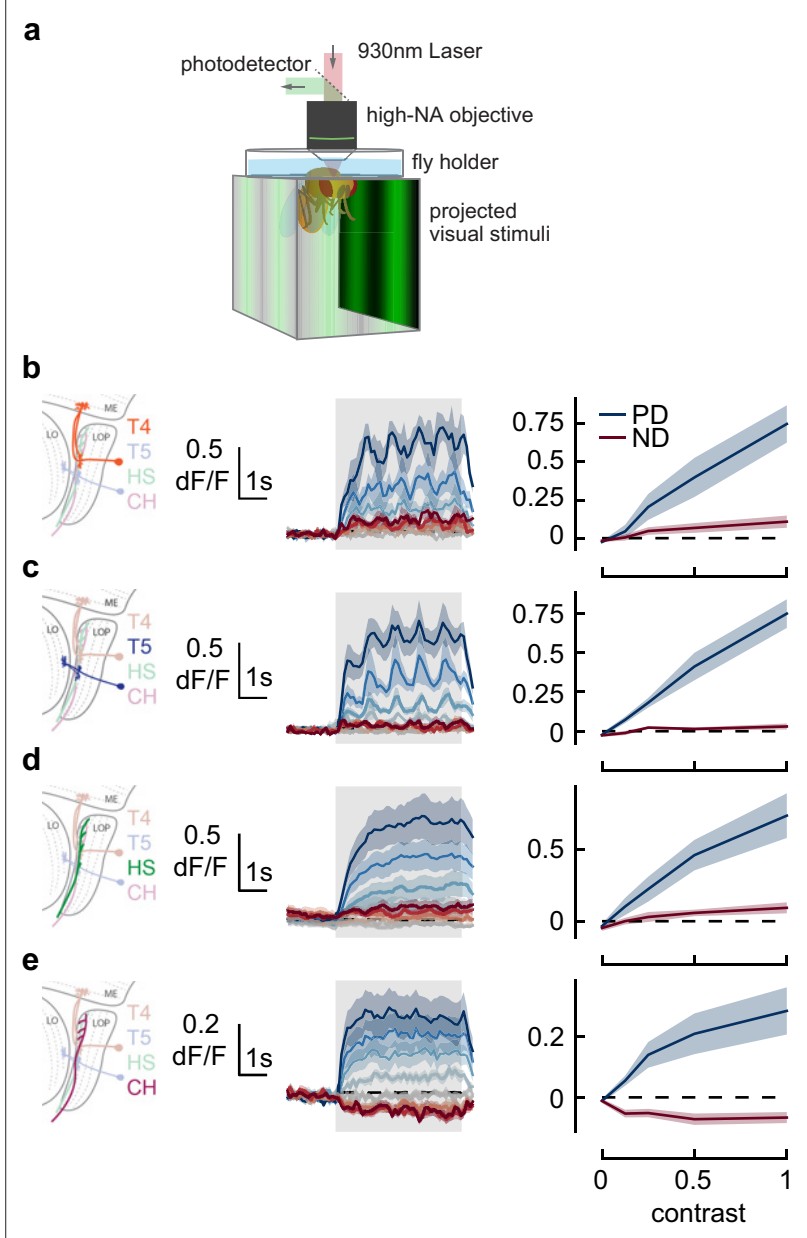

**Figure 5.** Responses in early direction-selective cells do not show a reduction or reversal of response on the timescale of the behavior. (**a**) We used two-photon microscopy to measure calcium activity in lobula plate neurons while presenting sinusoidal gratings at a range of contrasts. (**b**) T4 cells, marked in orange (*left*), responded to drifting sinusoidal gratings with increased calcium activity (*middle*). Darker colors indicate higher contrast, preferred direction in blue, null direction in red. When integrated across the stimulus presentation (*right*), calcium activity increased with stimulus contrast. N=8 flies. (**c–e**) As in (**b**) measuring calcium activity in T5, horizontal system (HS), and CH neurons. N=8, 10, 15 flies.

The online version of this article includes the following figure supplement(s) for figure 5:

**Figure supplement 1.** Anti-directional turning behavior occurs when using the optical filters also employed in the two-photon imaging experiments.

at various stages during early adulthood (**Figure 6**). Strikingly, as flies aged from 0.5 to 4 days post eclosion (dpe), the initial syn-directional turning became less transient and more sustained, indicative of a weaker anti-directional turning drive. We then wondered whether this plasticity was intrinsically programmed, or dependent on visual input. To disambiguate these possibilities, we reared flies in

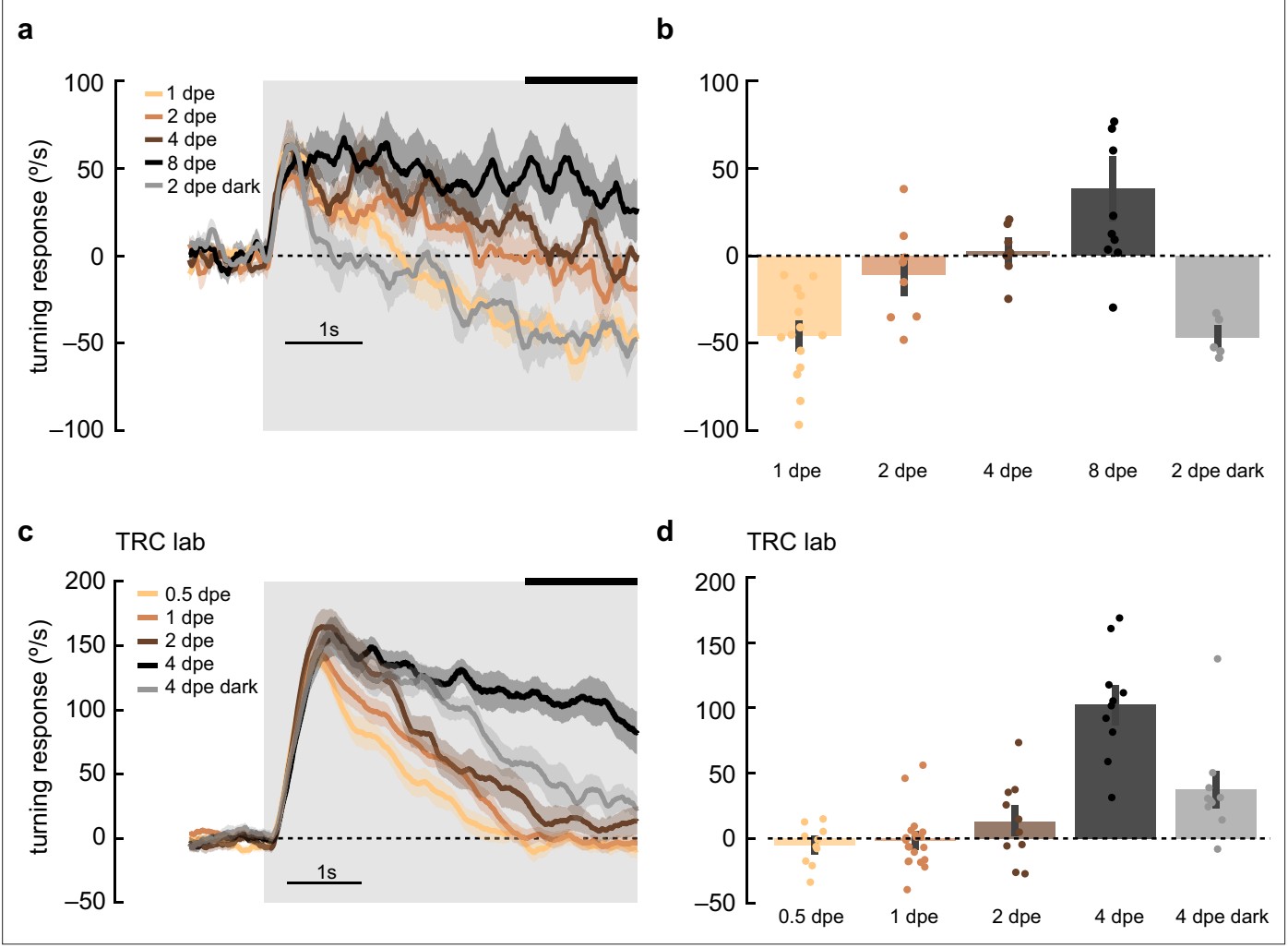

**Figure 6.** Maturation of optomotor response in early adulthood. (**a**) Adult flies at various ages post eclosion were presented with 5 s, high-contrast, rotating sinusoidal gratings as in *Figure 2b*. As the flies aged from 1 day post eclosion (dpe) to 2, 4, and 8 dpe, the initial anti-directional turning response transitioned into syn-directional turning. Dark-rearing flies at 2 dpe reduced this maturation effect. Shaded patches represent ±1 SEM. N=5–14 flies. (**b**) The last 1.5 s of the mean turning velocity of each fly was averaged, and the population response was plotted. (**c**) As in (**a**) but in the TRC lab, using 0.5, 1, 2, and 4 dpe, with dark rearing for 4 dpe. With maturation, the syn-directional turning became less transient. N=9–15 flies. (**d**) As in (**b**) but for data in (**c**).

The online version of this article includes the following figure supplement(s) for figure 6:

**Figure supplement 1.** *D. yakuba* lacks plasticity of anti-directional responses in adulthood that is observed *D. melanogaster*.

**Figure supplement 2.** Schematic of potential subtractive circuit properties consistent with data presented in this study.

darkness to 2 or 4 dpe and measured their turning responses (*Figure 6*, *gray*). Dark-reared flies exhibited a stronger deceleration away from syn-directional turning, similar to that found in more juvenile flies, arguing that visual input may sculpt the balance of syn- and anti-directional turning. Finally, we examined whether optomotor response plasticity could be detected in *D. yakuba*. However, in this species, anti-directional responses were stable across the first 4 days of adulthood, arguing that the role of visual experience in shaping these responses is itself evolutionarily tuned in drosophilids (*Figure 6—figure supplement 1*).

## Discussion

In this study, we found we could elicit robust turning in the opposite direction of high-contrast motion stimuli (*Figure 1*). This behavior is qualitatively different from other turning behaviors reported in the

literature (*Figures 2 and 3*), but shares elements with the circuitry necessary for optomotor behavior (*Figure 4*). However, the switch from syn-directional turning behavior to anti-directional turning behavior is not a reflection of changes in the activity of known direction-selective neuron types in the early visual system (*Figure 5*). Moreover, this anti-directional turning behavior exhibits a degree of experience-dependent plasticity (*Figure 6*).

## Anti-directional turning is distinct from other against-stimuli behaviors

The anti-directional turning behavior we have characterized is distinct from previous reports of flies turning in the direction opposite to the stimulus motion. First, some opposite-direction turning behaviors can be explained by stimulus aliasing (*Buchner, 1976*). Aliasing cannot explain our results because the stimulus that maximally activates anti-directional behavior has a spatial frequency of 1/60 cycles per degree, well below the Nyquist frequency of the fly eye (~1/10 cycles per degree) (*Götz, 1970*; *Buchner, 1976*) and below reports of higher acuity vision in flies (*Juusola et al., 2017*). Aliasing would also not explain the dependence on stimulus contrast.

Second, our observations also cannot be explained by stimuli to the rear of the fly driving it in the opposite direction (*Tammero et al., 2004*), since we observe anti-directional turning even when stimuli are only presented in only the 90° in front of the fly (*Figure 3*).

Third, it is also distinct from previous reports of reverse body saccades (*Williamson et al., 2018*) since it manifests in prolonged turns in the opposite direction of the stimulus and can be measured even when the magnitude of the turns is discarded (*Figure 3*).

Fourth, the behavior observed here also appears to be distinct from previously observed stimulus-density-dependent behavioral reversals (*Katsov and Clandinin, 2008*). Those previously reported behaviors showed immediate reversals, but it took ~1–2 s for flies in our paradigm to switch between optomotor and anti-directional behaviors.

## Anti-directional turning is unlikely to be due to adaptation to contrast alone

In mammalian retina, the direction preference of cells can switch because of upstream circuit adaptation (*Rivlin-Etzion et al., 2012*; *Vlasits et al., 2014*). However, we do not believe the anti-directional turning we observe has similar causes. In the mammalian retina, direction switching occurs when non-direction-selective neurons adapt to high-contrast stimuli, which distorts the downstream direction-selective computation. Since the adaptation in those experiments occurs in non-direction-selective neurons, it cannot be affected by the direction of the adapter stimulus. However, we see differences in turning behavior depending on whether we adapt with front-to-back or back-to-front stimuli (*Figure 2e*). This observation rules out a mechanism based solely on contrast, since the contrast content of front-to-back and back-to-front stimuli are identical.

The fly's visual system, however, adapts its gain to stimulus contrast (*Drews et al., 2020*; *Matulis et al., 2020*). Importantly, the phenomenology of the anti-directional turning also argues that the contrast adaptation is incomplete or heterogeneous among neurons, since contrast 1 and contrast 0.25 stimuli result in such different behaviors. Contrast adaptation reported in the fly is also faster than the 1–2 s preceding the shift to anti-directional turning in these experiments.

## Anti-directional turning behavior may require specific experimental and rearing conditions

Despite these previous reports of anti-directional turning under certain conditions, other labs have measured sustained optomotor turning in response to high-contrast stimuli (*Götz and Wenking, 1973*; *Seelig et al., 2010*; *Bosch et al., 2015*; *Strother et al., 2017*). Two major causes of this difference are likely display brightness and rearing conditions. Some experiments employ displays with mean luminances less than 5 cd/m$^2$ (*Rister et al., 2007*; *Seelig et al., 2010*; *Strother et al., 2017*). Our screens, with a mean luminance of 100 cd/m$^2$, are substantially brighter, but not especially bright when compared to natural scenes. In daytime natural scenes, foliage and the ground have average luminances of 200–500 cd/m$^2$ and the sky has an average luminance of around 4000 cd/m$^2$ (*Frazor and Geisler, 2006*). We suspect that as researchers move to using displays that can more accurately depict natural scene luminances, anti-directional turning behaviors will be encountered more frequently.

Rearing conditions also significantly influenced anti-directional turning behavior. Flies reared at 25°C showed less anti-directional behavior than those reared at 20°C. Temperature has known developmental effects on neural connectivity (*Kiral et al., 2021*). We also found differences based on fly age and fly strain. Notably, all three of these parameters vary significantly across the field, with prior studies varying rearing temperatures from 18–20°C to 25°C (see for instance *Juusola et al., 2017*; *Mongeau and Frye, 2017*; *Strother et al., 2017*; *Creamer et al., 2018*; *Ketkar et al., 2020*), varying ages from 1 day to 10 days (see for instance *Tammero et al., 2004*; *Bahl et al., 2013*; *Silies et al., 2013*), and varying strain between Canton-S or Oregon-R (see for instance *Rister et al., 2007*; *Clark et al., 2011*). Thus, these factors likely contribute to this phenomenon not having been previously reported, even as rotating sinusoids have been widely used in behavioral experiments.

## Tuning of anti-directional turning matches tuning of direction-selective neurons

The study of anti-directional turning behavior may yield clues about the temporal tuning of fly motion detectors. Optomotor behavior is tuned to visual stimuli in the range of 8–22 Hz (*Tammero et al., 2004*; *Tuthill et al., 2013*; *Creamer et al., 2018*; *Strother et al., 2018*), while anti-directional behavior is tuned to stimuli in the range of 0.5–4 Hz (*Figure 2*). Intriguingly, this slower tuning matches the tuning of T4, T5, and HS neurons, as measured via calcium imaging or electrophysiology (*Joesch et al., 2008*; *Chiappe et al., 2010*; *Maisak et al., 2013*; *Creamer et al., 2018*). Previous studies have suggested that the difference in tuning between behavior and imaging are due to octopamine that is released during behavior but not necessarily released during imaging (*Chiappe et al., 2010*; *Arenz et al., 2017*; *Strother et al., 2018*). In this work, we demonstrate a motion-related behavior tuned to low frequencies, comparable to those in neural measurements, during behavior that requires T4 and T5 neurons. Overall, this suggests that T4 and T5 are required for behaviors with very different temporal tuning, which in turn suggests that the temporal tuning of behavior is not determined solely by T4 and T5 tuning, but by other, parallel pathways as well.

## Anti-directional turning reveals circuits that turn the fly counter to visual motion

Experiments that show a decrease of turning over time to high-contrast stimuli (e.g., *Figure 1d*) could plausibly be explained by some kind of gain reduction or adaptation over time. However, the existence of turning in the direction opposite the stimulus motion in *D. melanogaster* and in *D. yakuba* requires a different explanation. These experiments reveal that over long timescales, a circuit that opposes the syn-directional optomotor turning response can dominate the behavioral response. Thus, this circuit is not simply scaling the magnitude of turning responses, but rather must be implementing an antagonistic, subtractive operation. We have schematized one possible subtractive circuit (*Figure 6—figure supplement 2*), which shows an influence by regressive motion detectors on long timescales that drives anti-directional turning. Measurements of free walking behavior have shown that the time constant of the autocorrelation of fly turning is around 100 ms (*Katsov et al., 2017*; *DeAngelis et al., 2019*). Opposing syn- and anti-directional turning circuits could be used to balance and tune the strength of turning responses on short timescales, while the anti-directional turning is revealed on longer timescales. This sort of subtractive processing predominates in computing motion signals in the visual systems of insects (*Mauss et al., 2015*) and mammals (*Rust et al., 2006*), and could also explain the existence of syn- and anti-directional turning behaviors.

In summary, we have presented evidence of a transition from syn-directional turning to no turning or to anti-directional turning when high-contrast stimuli are presented to the fly. This persists across laboratory environments and across *Drosophila* species and shows plasticity with age. This behavior suggests that turning in response to rotational stimuli is not a simple reflex. Instead, the turning is likely driven by circuits with opposing influences on turning direction. These circuits appear to differentially adapt to the direction and contrast of the stimulus. This complexity makes the optomotor response a model for studying the interactions of circuits as they control the low-dimensional behaviors that change an animal's orientation.

**Table 1.** Parental stock genotypes.

| Name | Genotype | Source | Stock # |
|------|----------|--------|---------|
| Wildtype | +; +; + (IsoD1) | *Gohl et al., 2011* | N/A |
| T4T5-Gal4 | +; +; R42F06-Gal4 (IsoD1 background) | BDSC | BDSC 41253 |
| HS-Gal4 | +; +; R27B03-Gal4 (IsoD1 bg) | *Seelig et al., 2010* | BDSC 49211 |
| CH-Gal4 | w; +; R35A10-Gal4 (Janelia bg) | BDSC | BDSC 49897 |
| UAS-Shibire^ts | +; +; UAS-Shibire^ts (IsoD1 bg) | *Silies et al., 2013* | N/A |
| Empty Gal4 | w; +; pBDPGAL4.1Uw (Janelia bg) | BDSC | BDSC 68384 |
| GCaMP6f | w; UAS-GCaMP6f; + | BDSC | BDSC 42747 |
| jGCaMP7b | w; +; UAS-jGCaMP7b | BDSC | BDSC 79029 |
| mtdTomato | w; +; UAS-mtdTomato | BDSC | BDSC 30124 |

## Methods

### Fly strains

Strains used in these experiments are listed in (*Tables 1 and 2*).

Genotypes of files used in imaging experiments: +; +; HS-Gal4/UAS-jGCaMP7b, +; UAS-GC6f/+; T4T5-Gal4/UAS-mtdTomato, w/+; +; CH-Gal4/UAS-jGCaMP7b.

### Fly rearing (DAC lab)

Unless otherwise noted, flies were reared at 20°C in Panasonic MIR-154-PA incubators (Panasonic/PHC, Tokyo, Japan). The flies were circadian entrained on 12 hr light-dark cycles. Flies were raised on Archon Scientific glucose food (recipe D20102, Archon Scientific, Durham, NC, USA). We used $CO_2$ to anesthetize flies more than 12 hr before the behavioral experiments.

Flies were tested for behavior in rigs built in the labs of DAC and TRC. Behavior shown in *Figure 1d, e*, *Figure 6c, d*, *Figure 1—figure supplement 1*, and *Figure 6—figure supplement 1* was acquired in the lab of TRC, while the rest was obtained in the lab of DAC.

### Fly rearing (TRC lab)

Flies were reared at 25°C, on molasses-based food, and circadian entrained on 12 hr light-dark cycles. Flies were collected within 3 hr of eclosion using brief $CO_2$ anesthetization. *D. melanogaster* and *D. yakuba* were raised under identical conditions. Dark-reared flies were put in a dark chamber within 3 hr of eclosion. Flies tested at 0.5 dpe were collected during the first 2 hr of the light cycle and were exposed to light until they were tested.

### Stimulus generation and behavioral turning assays (DAC lab)

Stimuli were presented using DLP Lightcrafter (Texas Instruments, Dallas, TX, USA) projectors (*Creamer et al., 2019*). Mirrors were used to bounce the projected light onto three screens made of back-projection material, surrounding the fly. The screens covered the front 270° around the

**Table 2.** Genotypes of flies used in behavior experiments.

| Experimental | Gal4 control | UAS control | Background control |
|------|------|------|------|
| T4T5-Gal4 x UAS- Shibire^ts: +; +; R42F06-Gal4/UAS-Shibire^ts | T4T5-Gal4 x IsoD1: +;+;R42F06-Gal4/+ | IsoD1 x UAS-Shibire^ts: +; +; +/UAS-Shibire^ts | IsoD1: +; +; + |
| HS-Gal4 x UAS- Shibire^ts: +; +; R27B03-Gal4/UAS-Shibire^ts | HS-Gal4 x IsoD1: +; +; R27B03-Gal4/+ | IsoD1 x UAS-Shibire^ts: +; +; +/UAS-Shibire^ts | IsoD1: +; +; + |
| CH-Gal4 x UAS-Shibire^ts: w/+; +; R35A10-Gal4/UAS-Shibire^ts | CH-Gal4 x IsoD1: w/+; +; R35A10-Gal4/+ | Empty Gal4 x UAS-Shibire^ts: +/w; +; pBDPGAL4.1Uw /UAS-Shibire^ts | Empty Gal4 X IsoD1: +/w; +; +/pBDPGAL4.1Uw |

fly, and ~45° in elevation above and below the fly. The projectors were set to monochrome mode (green unless otherwise noted), updating at 180 Hz. Stimulus video was generated through a custom MATLAB (Mathworks, Natick, MA, USA) application using PsychToolbox (*Kleiner et al., 2007*). Stimuli were mapped onto a virtual cylinder around the fly and the MATLAB application generated a viewpoint-corrected video signal. Stimuli are as described in the text. Unless stated otherwise, stimuli were preceded and succeeded by uniform intensity stimuli with the same mean luminance; in our figures, this is represented by gray periods. These interleaved uniform intensity stimuli typically lasted 5 s.

Behavioral experiments were performed 12–60 hr after staging. For behavioral experiments, we selected female flies, and co-housed them with males after staging. Flies were cold-anesthetized and fixed to needles using UV-cured epoxy (Norland optical adhesive #63, Norland Products, Cranbury, NJ, USA). Flies were then placed above air-suspended polypropylene balls in a dark box maintained at 34–36°C in a room held at ~50% relative humidity (such that the box has relative humidity of 25–30%). These conditions yield robust walking and turning behaviors. These balls were 6 mm in diameter and weighed ~120 mg. The balls were painted with two layers of marker coatings – a base silver layer and a red top layer. The motion of balls was detected by either a Parallax mouse sensor board (Parallax, Rocklin, CA, USA) with an MCS-12086 sensor (Unity Opto Technology, Taipei, Taiwan), or a custom board with an ADNS 2080 sensor (Avago Technologies/Broadcom Inc, San Jose, TX, USA). The data from these sensors were transferred to a custom MATLAB application via an Arduino Uno board. Flies were excluded from analysis if they walked at an average speed of <1 mm/s over the entire experiment, or if the standard deviation of their turning was less than 1.5 times their meaning turning, an indication that they turned strongly and constitutively in one direction. These criteria typically excluded <10% of flies run.

## Stimulus generation and behavioral turning assays (TRC lab)

Stimuli were presented using a DLP Lightcrafter (Texas Instruments, Dallas, TX, USA) projector. Three coherent optic fibers were used to direct the projected light onto three screens made of back-projection material, surrounding the fly (*Clark et al., 2011*; *Clark et al., 2014*). The screens covered the front 270° around the fly, and ~45° in elevation above and below the fly. The projectors were set to monochrome mode, updating at 120 Hz. Stimulus video was generated through Flystim (*Choi et al., 2023*), a custom Python application developed in the Clandinin Lab (*Turner et al., 2022*). Stimuli were mapped onto a virtual cylinder around the fly and Flystim generated a viewpoint-corrected video signal. Stimuli are as described in the text. Stimuli were preceded and succeeded by uniform intensity stimuli with the same mean luminance; in our figures, this is represented by the gray periods. These interleaved uniform intensity stimuli typically lasted 5 s.

Behavioral experiments were performed 12–48 hr after eclosion, as described in the figures. Flies were cold-anesthetized and fixed to needles using UV-cured adhesive (Bondic, Niagara Falls, NY, USA). Flies were then placed above air-suspended balls made with LAST-A-FOAM FR-4615 polyurethane foam (General Plastics, Tacoma, WA, USA). These balls were 9 mm in diameter and weighed ~91.7 mg. The motion of balls was detected by a Flea3 FL3-U3-13Y3M camera (Teledyne Flir, Wilsonville, OR, USA) and Fictrac software (*Moore et al., 2014*). The entire apparatus was contained in a light-tight box maintained at 34°C in a room held at ~50% relative humidity (such that the box has relative humidity of 25–30%). These conditions yield robust walking and turning behaviors. There were no exclusion criteria applied to flies in analyzing these experiments.

## Luminance measurements

Our experiments revealed that luminance was a critical factor in generating the observed anti-directional turning. In the rigs in the DAC and TRC labs, we measured the luminance by placing a power meter in front of the rear-projection screens. The total power was converted to luminous intensity by dividing by the area of the sensor and by $2\pi$ steradians, approximating the scattering by the screen as uniform in all directions. We then converted from watts to candelas by the standard factor and weighting by the photopic luminosity function centered on our mean screen wavelength (~540 nm). Though the human photopic luminosity function is an unusual choice for studying fly visual behavior, $cd/m^2$ has been a standard unit for measuring screen luminous intensity in the field.

## Imaging procedures

Two-photon imaging (*Figure 5*) was performed as previously described (*Tanaka and Clark, 2022*). Briefly, two-photon images were acquired with a Scientifica microscope at between 6 Hz and 13 Hz using a 930 nm femtosecond laser (SpectraPhysics, Santa Clara, CA, USA) using ScanImage (*Pologruto et al., 2003*). Visual stimuli were presented on three screens occupying 270° of azimuthal angle about the fly using projectors (*Creamer et al., 2019*). Optical filters on the projector and emission filters prevented the visual stimulus light from leaking into the two-photon images.

Regions of interest (ROIs) were extracted from image timeseries using a watershed algorithm. Responsive ROIs were included in the analyses. For T4 and T5 neurons, each ROI was identified as a T4-dominant or T5-dominant ROI by its response to light vs. dark edges, following prior procedures (*Agrochao et al., 2020*). For all neuron types, responses were averaged over ROIs and over trials of each stimulus type to obtain a measurement for each fly; these fly measurements acted as the independent measurements to compute means and standard error bars for the figure.

## Statistical tests

Throughout the paper, each fly was considered an independent sample for statistical purposes. Means and standard errors were computed over flies. For imaging experiments, ROIs from a specific neuron type were first averaged within each fly, creating a value for each fly's response. These values were used to calculate means and standard errors over the tested flies. In the silencing experiments, a two-sample Student's t-test was used to test for significant differences between the experimental genotype and parental controls.

## Acknowledgements

This work was supported by NIH R01EY026555 (DAC), R01EY022638 (TRC), and by a Chan-Zuckerberg Investigator Award (TRC). MC was supported by a NDSEG Fellowship and a Regina Casper Stanford Graduate Fellowship; RT was supported by the Takenaka Foundation; MSC was supported by the NSF GRFP; NCBM was supported by a CAPES Fellowship; JS was supported by a Ford Foundation Fellowship. We also gratefully acknowledge Irving E Wang, who performed initial experiments examining optomotor responses in many drosophilid species in the lab of TRC.

## Additional information

### Competing interests

Damon A Clark: Reviewing editor, *eLife*. The other authors declare that no competing interests exist.

### Funding

| Funder | Grant reference number | Author |
|---|---|---|
| National Institutes of Health | R01EY026555 | Damon A Clark |
| National Institutes of Health | R01EY022638 | Thomas R Clandinin |
| Chan Zuckerberg Initiative | Investigator Award | Thomas R Clandinin |
| National Defense Science & Engineering Graduate | | Minseung Choi |
| Stanford University | Regina Casper Stanford Graduate Fellowship | Minseung Choi |
| Takenaka Foundation | | Ryosuke Tanaka |
| National Science Foundation | National Science Foundation Graduate Research Fellowship Program | Matthew S Creamer |

| Funder | Grant reference number | Author |
|---|---|---|
| CAPES Foundation | Fellowship | Natalia CB Matos |
| Ford Foundation | Fellowship | Joseph W Shomar |

The funders had no role in study design, data collection and interpretation, or the decision to submit the work for publication.

## Author contributions

Omer Mano, Conceptualization, Data curation, Software, Formal analysis, Investigation, Visualization, Writing - original draft, Writing - review and editing; Minseung Choi, Conceptualization, Data curation, Software, Visualization, Methodology, Writing - review and editing; Ryosuke Tanaka, Data curation, Formal analysis, Investigation; Matthew S Creamer, Data curation, Software, Formal analysis, Investigation; Natalia CB Matos, Joseph W Shomar, Bara A Badwan, Investigation; Thomas R Clandinin, Conceptualization, Supervision, Funding acquisition, Methodology, Project administration, Writing - review and editing; Damon A Clark, Conceptualization, Supervision, Funding acquisition, Visualization, Methodology, Writing - original draft, Project administration, Writing - review and editing

## Author ORCIDs

Omer Mano ⓘ http://orcid.org/0000-0003-2606-8567
Minseung Choi ⓘ http://orcid.org/0000-0001-5398-219X
Joseph W Shomar ⓘ http://orcid.org/0009-0000-9534-6263
Thomas R Clandinin ⓘ http://orcid.org/0000-0001-6277-6849
Damon A Clark ⓘ http://orcid.org/0000-0001-8487-700X

## Decision letter and Author response

Decision letter https://doi.org/10.7554/eLife.86076.sa1
Author response https://doi.org/10.7554/eLife.86076.sa2

## Additional files

### Supplementary files

• MDAR checklist

### Data availability

Raw behavioral and imaging data, along with code to run the analyses and create the plots in this paper, are available on Zenodo: https://doi.org/10.5281/zenodo.19424913.

The following dataset was generated:

| Author(s) | Year | Dataset title | Dataset URL | Database and Identifier |
|---|---|---|---|---|
| Mano O, Choi M, Tanaka R, Creamer M, Matos N, Shomar J, Badwan B, Clandinin T, Clark D | 2026 | Behavioral and 2p imaging data and analysis code: Long timescale anti-directional rotation in Drosophila optomotor behavior | https://doi.org/10.5281/zenodo.19424913 | Zenodo, 10.5281/zenodo.19424913 |

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
