## [Editor Report]

The present study provides a valuable new perspective on the optomotor response based on an inversion of the behavior under specific (non-natural) conditions that may help elucidate the principles of this specific behavior. The evidence provided is convincing.

---

## [Decision Letter]

**Decision letter after peer review:**

Thank you for submitting your article "Long timescale anti-directional rotation in *Drosophila* optomotor behavior" for consideration by *eLife*. Your article has been reviewed by 3 peer reviewers, one of whom is a member of our Board of Reviewing Editors, and the evaluation has been overseen by Claude Desplan as the Senior Editor.

Essential revisions:

As you can see below, there was a consensus that the observed anti-directional turning is an interesting behavior that may help understand principles of optomotor behavior. However, it was less clear to the reviewers how this 'non-natural' behavior precisely informs about normal behavior and what the implications may be for a broader audience. An effort by the authors in clarifying these points might improve the paper.

In addition, the following revisions suggested by reviewer 2 were deemed essential to strengthen the conclusions such as they are:

1) Given the dynamics of the behavior, it is important to look at the turning dynamics after the stimulus has stopped. If direction-selective adaptation mechanisms are regulating the turning response, one may find long-lasting biases even in the absence of stimulation. If the authors have more data after the stimulus end, it would be good to further expand the time range by a few seconds to show if this is the case or not (for example, in Figure 1b).

2) Another important set of experiments is to perform closed-loop configuration tests that should allow the motion computing circuitry to adapt to the chosen environmental conditions. Explorations of the changes in turning response dynamics after such treatments should then enable further dissections of the mechanisms of adaptation. Closed-loop experiments under different contrast conditions have already been performed (for example, Leonhardt et al. 2016), which also showed complex response dynamics after stimulus on- and offset.

*Reviewer #3 (Recommendations for the authors):*

In Figure 3, noting that flies could turn with both fast and slow movements, which may be in different directions, and information may be lost in the average trace, the authors show the raw turning traces, as well as a binarized version that shows only turning direction. It is nice to see the raw traces, but I think it could also be helpful to the reader to summarize this data by showing the distribution of angular velocities at different time points.

The authors may want to cite the following work which describes anti-directional optomotor behavior in fish.

Bak-Coleman, J., Smith, D., and Coombs, S. (2015). Animal Behaviour. Animal Behaviour, 107(C), 7-17. http://doi.org/10.1016/j.anbehav.2015.06.007

---

## [Author Response]

Essential revisions:As you can see below, there was a consensus that the observed anti-directional turning is an interesting behavior that may help understand principles of optomotor behavior. However, it was less clear to the reviewers how this 'non-natural' behavior precisely informs about normal behavior and what the implications may be for a broader audience. An effort by the authors in clarifying these points might improve the paper.In addition, the following revisions suggested by reviewer 2 were deemed essential to strengthen the conclusions such as they are:1) Given the dynamics of the behavior, it is important to look at the turning dynamics after the stimulus has stopped. If direction-selective adaptation mechanisms are regulating the turning response, one may find long-lasting biases even in the absence of stimulation. If the authors have more data after the stimulus end, it would be good to further expand the time range by a few seconds to show if this is the case or not (for example, in Figure 1b).

We agree with the reviewer that this is an important detail that we did not show in the original submission. In the revised submission, we have expanded the time trace in Figure 1b to show 5 seconds after the stimulus has ended. The dynamics after high and low contrast stimuli are unremarkable, with each trace returning gradually to a baseline of zero net turning.

2) Another important set of experiments is to perform closed-loop configuration tests that should allow the motion computing circuitry to adapt to the chosen environmental conditions. Explorations of the changes in turning response dynamics after such treatments should then enable further dissections of the mechanisms of adaptation. Closed-loop experiments under different contrast conditions have already been performed (for example, Leonhardt et al. 2016), which also showed complex response dynamics after stimulus on- and offset.

Thank you for this suggestion. We have performed this experiment, following the more detailed suggestion in Reviewer #2’s comments. As such, we presented high contrast sinusoid gratings in closed-loop for 5 seconds with a gain of 1 (i.e., 90º change in ball orientation yields 90º change in stimulus orientation), following which we presented the standard 5 seconds of high contrast, open-loop rotation. In these experiments, we also included a positive control of a gray screen preceding the open-loop rotation, which should yield anti-directional turning, and a stationary, high-contrast grating that lasts for 5 seconds before the open-loop stimulus. As expected, the positive control worked, and the stationary grating adapter yielded almost identical turning to the mean-gray adapter. We found that the response to the closed-loop stimulus looked virtually identical to the parallel open-loop control cases. The closed-loop measurements were noisier because our rig presents the same stimulus to 5 flies simultaneously; our closed loop conditions place each fly in control of the stimulus for 1/5 of the experiment duration, resulting in fewer closedloop trials per fly.

In the revised manuscript, the results of these experiments are shown in Figure 2 – Supp. Figure 1 and discussed in the results near the Figure 2 adaptation results. We now also mention in our paper that data in Leonhardt shows closed-loop responses under different contrast conditions and use that to motivate our new test of closed-loop adaptation conditions.

Reviewer #3 (Recommendations for the authors):In Figure 3, noting that flies could turn with both fast and slow movements, which may be in different directions, and information may be lost in the average trace, the authors show the raw turning traces, as well as a binarized version that shows only turning direction. It is nice to see the raw traces, but I think it could also be helpful to the reader to summarize this data by showing the distribution of angular velocities at different time points.

Thank you for this suggestion. We have added a plot showing the rotational velocity distribution in individual trials over time, now shown in Figure 3 – Supp. Figure 1. We are not sure that it particularly illuminates the structure of the turning behavior, though saccadelike behavior is not obviously visible in these distributions. The new plot does give an accurate sense of the variability in individual trials, and shows that the mean seems like a reasonable summary statistic of the distribution.

The authors may want to cite the following work which describes anti-directional optomotor behavior in fish.Bak-Coleman, J., Smith, D., and Coombs, S. (2015). Animal Behaviour. Animal Behaviour, 107(C), 7-17. http://doi.org/10.1016/j.anbehav.2015.06.007

Thank you for this citation! We were unaware of this work, but have now added it to our introduction where we talk about previously reported anti-direction turning phenomenology.